# Profiling the inhibitory receptors LAG-3, TIM-3, and TIGIT in renal cell carcinoma reveals malignancy

Kimiharu Takamatsu [1], Nobuyuki Tanaka [1✉], Kyohei Hakozaki[1], Ryohei Takahashi[1], Yu Teranishi[1], Tetsushi Murakami[1], Ryohei Kufukihara[1], Naoya Niwa[1], Shuji Mikami[2], Toshiaki Shinojima[1,3], Takashi Sasaki [4], Yusuke Sato[5], Haruki Kume[5], Seishi Ogawa [6], Kazuhiro Kakimi[7], Takashi Kamatani[8,9,10], Fuyuki Miya [9], Tatsuhiko Tsunoda[8,9,11], Eriko Aimono[12], Hiroshi Nishihara[12], Kazuaki Sawada[13], Takeshi Imamura[14,15], Ryuichi Mizuno[1] & Mototsugu Oya [1]

A cutting edge therapy for future immuno-oncology is targeting a new series of inhibitory receptors (IRs): LAG-3, TIM-3, and TIGIT. Both immunogenomic analyses and diagnostic platforms to distinguish candidates and predict good responders to these IR-related agents are vital in clinical pathology. By applying an automated single-cell count for immunolabelled LAG-3, TIM-3, and TIGIT, we reveal that individual IR levels with exclusive domination in each tumour can serve as valid biomarkers for profiling human renal cell carcinoma (RCC). We uncover the immunogenomic landscape associated with individual IR levels in human RCC tumours with metastases in various organs and histological subtypes. We then externally validate our results and devise a workflow with optimal biomarker cut-offs for discriminating the LAG-3, TIM-3, and TIGIT tumour profiles. The discrimination of LAG-3, TIM-3, and TIGIT profiles in tumours may have a broad impact on investigations of immunotherapy responses after targeting a new series of IRs.

[1] Department of Urology, Keio University School of Medicine, 160-8582 Tokyo, Japan. [2] Department of Diagnostic Pathology, Keio University Hospital, 160-8582 Tokyo, Japan. [3] Department of Urology, Saitama Medical University, Moroyama, Saitama, Japan. [4] Center for Supercentenarian Medical Research, Keio University School of Medicine, Tokyo, Japan. [5] Department of Urology, Graduate School of Medicine, The University of Tokyo, Tokyo, Japan. [6] Department of Pathology and Tumour Biology, Graduate School of Medicine, Kyoto University, Kyoto, Japan. [7] Department of Immunotherapeutics, The University of Tokyo Hospital, Tokyo, Japan. [8] Laboratory for Medical Science Mathematics, Department of Biological Sciences, Graduate School of Science, The University of Tokyo, Tokyo, Japan. [9] Department of Medical Science Mathematics, Medical Research Institute, Tokyo Medical and Dental University (TMDU), Tokyo, Japan. [10] Department of Pulmonary Medicine, Keio University School of Medicine, Tokyo, Japan. [11] Laboratory for Medical Science Mathematics, RIKEN Center for Integrative Medical Sciences, Yokohama, Japan. [12] Genomics Unit, Keio Cancer Center, Keio University School of Medicine, Tokyo, Japan. [13] Center for Integrated Medical Research, Keio University School of Medicine, Tokyo, Japan. [14] Department of Molecular Medicine for Pathogenesis, Graduate School of Medicine, Ehime University, Toon, Japan. [15] Translational Research Center, Ehime University Hospital, Toon, Japan. ✉email: urotanaka@keio.jp

The clinical success of the initial inhibitory receptors (IRs) CTLA-4, PD-1 or PD-L1 in cancers remains fresh in our minds. Recent advances in medicine have shown that ipilimumab, which was first approved for skin melanoma, targets CTLA-4[1]; later, PD-1–targeting nivolumab/pembrolizumab was approved for cancers of the skin[2–4], lung[5,6], and kidney[7,8]. With the substantial survival benefits offered by nivolumab, renal cell carcinoma (RCC) has become one of hallmarks for immuno-oncology treatment in this decade[9–12]. The clinical efficacy achieved by the combination of ipilimumab and nivolumab for RCC has been encouraging for both patients and clinicians[13,14], as has the combination of anti-angiogenic axitinib with pembrolizumab and avelumab[15,16]. However, a substantial proportion of patients remain refractory or have developed acquired resistance to the first series of immunotherapy regimens (i.e. CTLA-4, PD-1 or PD-L1 monotherapies and their combinations)[11,17]. Thus, emerging demands for new therapies targeting other IRs have led to a broadened therapeutic repertoire in RCC and, more broadly, all cancer types.

To date, three new IRs have been spotlighted, leading to intensified competition in drug development: lymphocyte activation gene 3 (LAG-3), T-cell immunoglobulin and mucin domain 3 (TIM-3), and T-cell immunoreceptor with Ig and ITIM domains (TIGIT)[18–21]. LAG-3, also called CD223, is upregulated on stimulated T-cells and is an immune checkpoint for preventing excessive activation[22]. TIM-3 (CD366) was first identified as a central regulator of IFN-γ-secreting type 1 Th cells and immune tolerance[23]; later, however, TIM-3 was better characterized as an IR that controls both anti-viral immunity and anti-tumour immunity, similar to LAG-3[19]. TIGIT is a member of the Ig superfamily that behaves as a co-inhibitory receptor on the immune cell surface, typically binding ligands CD155 and CD112[18].

Targeting these IRs may soon become a reality as the second series of immunotherapies to be translated to the clinic. Therefore, the next question is whether targeting these new IRs with new monotherapies or in combination with existing drugs will achieve clinical efficacy. The answer to this central question is still unknown and requires further discussion, but the combination of IR-targeting agents with anti-PD-1/PD-L1 blockade may be promising for clinical use based on clinical trials[19,20]. Facing the new immuno-oncology era, together, we need to decide on the optimal combinations of treatment options for each patient, i.e. the second series of IRs (LAG-3, TIM-3, and TIGIT). What is the best biomarker for the selection of a second series of IR-related agents? Who can benefit from targeting these new IRs?

To identify good responders, it is vital to examine the expression levels of LAG-3, TIM-3, and TIGIT. We preliminarily found that clear cell RCC (ccRCC) cohorts could be immuno-histologically divided into three risk groups based on the dominant expression of the second series of IRs, namely, the LAG-3, TIM-3, and TIGIT clusters[24]. In this study, we first seek to expand on prior studies by using our immunogenomics dataset of primary ccRCCs and unravel the relationships among the three IR signatures, clinical outcomes, genetic alterations, and tumour immune microenvironment. Second, we aim to deepen our understanding of the second series of IRs by examining RCCs with metastases in various organs and different histological sub-types. Validation and subsequent biomarker cut-off selection are essential for future pathology. Third, by applying an external validation dataset, we strengthen our hypothesis and determine an optimal workflow and biomarker cut-off for discriminating the tumour profiles of LAG-3, TIM-3, and TIGIT by using clinically based immunohistochemistry.

## Results

**Automated single-cell pathology and clustering.** First, we sought to apply an automated single-cell count for immunolabelled LAG-3, TIM-3, and TIGIT to 105 primary ccRCC tumour samples (i.e. COHORT 1, Table 1). A quantitative immunohistological assessment can provide a robust research platform worldwide. In this automated analysis, positively stained immune cells in tumours were distinguished together with nuclei and counted, as illustrated for LAG-3, TIM-3, and TIGIT (Fig. 1a–c, respectively). The mean number of cells was $102.3 \pm 16.8/mm^2$ for

**Table 1 Clinicopathological characteristics associated with the phenotypic signatures of the new IRs (LAG-3, TIM-3, and TIGIT) in 105 primary ccRCC tumour samples (COHORT 1).**

| Characteristic | All patients (n = 105) | IR profile | | | p value | | |
| --- | --- | --- | --- | --- | --- | --- | --- |
| | | LAG-3 n = 45 (43%) | TIM-3 n = 46 (44%) | TIGIT n = 14 (13%) | LAG-3 vs TIM-3 | LAG-3 vs TIGIT | TIM-3 vs TIGIT |
| Age, yr, median, (IQR) | 60 (52-70) | 63 (55-73) | 57 (52-70) | 54 (50-68) | 0.756 | 0.231 | 0.322 |
| Gender, no (%): | | | | | 0.247 | 1.000 | 0.515 |
| Male | 76 (72%) | 35 (78%) | 30 (65%) | 11 (79%) | – | – | – |
| Female | 29 (28%) | 10 (22%) | 16 (35%) | 3 (21%) | – | – | – |
| Nuclear grade, no (%): | | | | | 0.668 | 0.109 | 0.192 |
| G1 + G2 | 69 (66%) | 27 (60%) | 30 (65%) | 12 (86%) | – | – | – |
| G3 + G4 | 36 (34%) | 18 (40%) | 16 (35%) | 2 (14%) | – | – | – |
| Pathological T stage, no (%): | | | | | 0.290 | 0.354 | 0.756 |
| pT1 + pT2 | 64 (61%) | 24 (53%) | 30 (65%) | 10 (71%) | – | – | – |
| pT3 + pT4 | 41 (39%) | 21 (47%) | 16 (35%) | 4 (29%) | – | – | – |
| Venous invasion, no (%): | | | | | 0.112 | 0.342 | 1.000 |
| Yes | 30 (29%) | 17 (38%) | 10 (22%) | 3 (21%) | – | – | – |
| No | 75 (71%) | 28 (62%) | 36 (78%) | 11 (79%) | – | – | – |
| Tumour size, mm, median (IQR) | 59 (35–80) | 50 (31–74) | 68 (49–81) | 48 (30–81) | 0.186 | 0.658 | 0.177 |

p values were determined with a two-tailed Mann–Whitney U-test or Fisher's exact test.

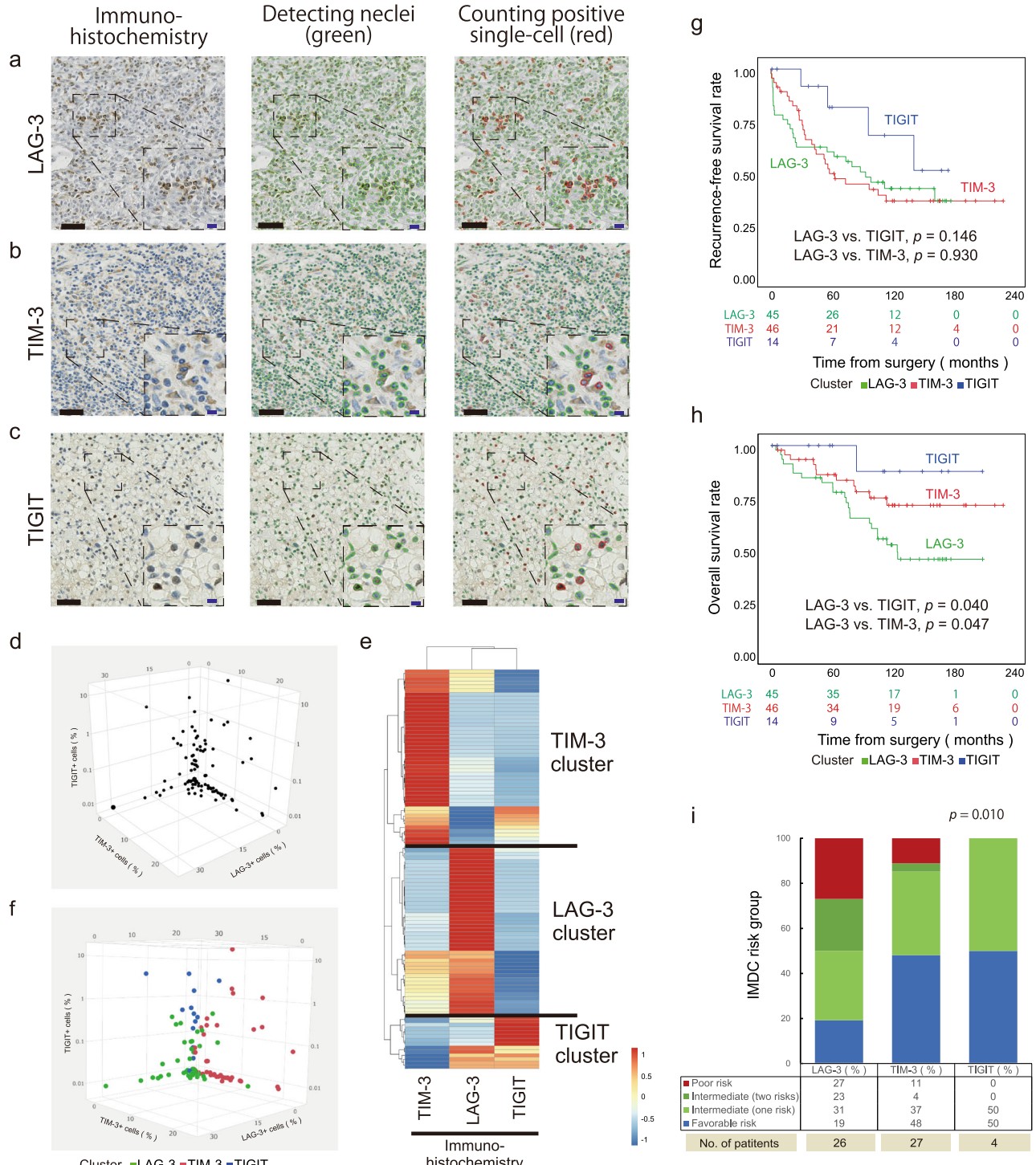

**Fig. 1 Identification and characterization of the phenotypic signatures of the new IRs (LAG-3, TIM-3, and TIGIT) in human ccRCC tumours. a–c**
Automated cell-by-cell segmentations and single-cell counts of human ccRCC tumours immunolabelled for LAG-3 (**a**), TIM-3 (**b**), and TIGIT (**c**). Zoomed-in images of the indicated boxed regions. Scale bars, 50 µm (black) and 20 µm (blue). The experiment was performed one time because of using human sample. **d** Three-dimensional plots showing positive-stained cell densities for LAG-3, TIM-3, and TIGIT in 105 ccRCC patients. **e** Hierarchical clustering heatmap (low, blue; high, red) using the positively stained cell densities of LAG-3, TIM-3, and TIGIT in (**d**). Data were normalized prior to clustering. **f** Labelling of three-dimensional plots by the inferred cluster types obtained in (**e**). **g**, **h** Kaplan–Meier survival curves for recurrence-free survival (**g**) and overall survival (**h**) following surgery in 105 ccRCC patients based on the inferred cluster types. *p* values were determined with log-rank test. **i** Percent distribution of the inferred cluster types within IMDC risk groups. *p* values were determined with a two-tailed Fisher's exact test (favourable and intermediate (one risk factor) risk groups vs. intermediate (two risk factors) and poor risk groups). Source data are provided as a Source data file.

LAG-3, $473.4 \pm 51.5/mm^2$ for TIM-3, and $6.0 \pm 1.8/mm^2$ for TIGIT. However, the total cell density may vary from region to region in individual tumours, so we then divided each receptor-positive cell density obtained to the total cell density from the same sample region. The results showed that the mean percentages of receptor-positive cell density were 3.3% for LAG-3, 4.5% for TIM-3, and 0.4% for TIGIT; these values were used for all analyses hereafter. Plotting each IR-positive cell density in three dimensions showed that distinct IR levels were present, with exclusive domination in each tumour allowing small fractions to overlap (Fig. 1d). Thus, we hypothesized that primary ccRCC profiles can be divided robustly with our automated platform based on the signatures of the three IRs.

To answer this question, we first assessed whether the expression of the three IRs was exclusive at the single-cell level by multi-colour staining of LAG-3, TIM-3, and TIGIT (Supplementary Fig 1a). Co-staining the three IRs and counting the individual IR-positive cells from a total of 104,236 cells in COHORT 1 samples revealed that most IR-positive cells expressed only one marker (Supplementary Fig 1b). Although small percentages of them overlapped (Supplementary Fig 1c), our results suggest that the cells in the tumour area can dominantly express one of the three IRs in ccRCC.

Next, by applying hierarchical clustering to the 105 ccRCC tumour samples based on the normalized LAG-3-, TIM-3-, and TIGIT-positive cell densities in each tumour, we successfully identified three groups with distinct IR levels (Fig. 1e). The phenotypic signatures of TIM-3 (44%), LAG-3 (43%) and TIGIT (13%) were individually dominant in independent clusters (Fig. 1f). Since there was no definitive tendency for pathological features among the three clusters (Table 1), a prognostic analysis of our 105 primary ccRCC tumours revealed the inner landscape of this cluster analysis. Although recurrence-free survival was not different among the three clusters (Fig. 1g, Supplementary Table 1), the LAG-3 cluster was associated with the worst overall survival rates in Kaplan–Meier analysis, showing significant differences with patients with favourable TIM-3 ($p = 0.047$) and TIGIT ($p = 0.040$) clusters (Fig. 1h). Furthermore, in multi-variable analysis, the LAG-3 cluster ($p = 0.037$) was an independent risk factor for overall mortality following surgery, along with patient age, nuclear grade, and tumour size in COHORT 1 (Supplementary Table 1, Supplementary Data 1).

Clinically, the International Metastatic RCC Database Consortium (IMDC) Risk Model, utilizing six readily available factors (time from surgery to systemic therapy, performance status, haemoglobin, calcium, neutrophil, and platelet counts), is a well-validated tool for ccRCC patients with metastases and correlates anti-angiogenic treatment and immunotherapy susceptibility[9,10,12]. In total, 57 patients (54%) experienced recurrence and were assigned to the appropriate risk category by the IMDC indicators: favourable (0 risk factor): 35%, intermediate (1 or 2 risk factors): 47%, and poor (3 risk factors or greater): 18%. We then studied the relationship between the IMDC risk criteria and the IR profiles in ccRCC. Interestingly, plotting the IR profiles against the IMDC indicators revealed a certain tendency (Fig. 1i): patients in the LAG-3 cluster had worse IMDC risk at disease relapse, implying the validity of our depicting the LAG-3 cluster as having the worse overall survival in Fig. 1h, demonstrating resistance to existing anti-angiogenic treatment and immunotherapy.

**Immunogenomic differences**. Next, we sought to examine the genomic alterations underlying the signatures of the three new IRs in ccRCC, since recent advances in sequencing have revealed a subset of genes that correlate with the response to anti-angiogenic therapy as well as immuno-oncologic therapy[25,26].

Herein, we analysed forty-three ccRCC tumour samples from COHORT 1 comprising the LAG-3 ($n = 16$), TIM-3 ($n = 19$), and TIGIT ($n = 8$) clusters for alterations in 160 cancer-associated genes (Fig. 2a). The most frequently altered genes across the three IR spectra (>5%) were *VHL*, *PBRM1*, *SETD2*, *MTOR*, *TP53*, and *ATM*, indicating that the incidence of VHL mutations in our cohort is slightly less than that in the previous studies[27]. Interestingly, genetic alterations of the p53/cell cycle pathway in the LAG-3 cluster were obvious in our population ($p = 0.020$, Fig. 2b).

We next investigated the association between the three new IR clusters and the tumour immune microenvironment. In total, fifteen immunolabelled molecules, including 6 for acquired immunity, 3 for innate immunity, 4 for cancer metabolism, and 2 for cancer stroma, were assessed by automated signal segmentation in the 105 ccRCC tumour samples from COHORT 1. Cell-by-cell immunohistological analysis for acquired immunity revealed that the LAG-3 cluster had higher levels of tumour-infiltrating CD8 T-cells than the two other clusters (Fig. 2c). Furthermore, tumours in the LAG-3 cluster were associated with high levels of CD39 in CD8 T-cells, revealing substantial cell exhaustion in tumour-infiltrating CD8 T-cells in these tumours (Fig. 2c)[28]. However, the CTLA-4 level was high in the TIGIT cluster.

Then, our immunohistological analysis for innate immunity revealed that the LAG-3 cluster had significantly higher levels of CD163 than the two other clusters (Fig. 2d), demonstrating that many infiltrating tumour-associated macrophages (TAMs) in tumours belong to the LAG-3 cluster (co-staining of CD68 together with CD163 from a total of 123,460 cells in COHORT 1 samples showed 84.5% concordance in 1163 CD163-positive cells and 983 CD163 and CD68 positive cells). Our assessments were further extended to the state of tumour metabolism in ccRCC samples labelled for the proliferative Ki-67, indoleamine 2,3-dioxygenase-1 (IDO-1), GLUT-1, and CD73 markers, but no difference was noted in inhibitory tumour metabolism among the three groups (Fig. 2e). Furthermore, two molecules associated with the cancer stroma, i.e., CD34 (to label blood vessels) and D2-40 (to label lymph vessels), were assessed, and we found that tumours in the LAG-3 cluster were less dependent on the development of lymph vessels than those in the TIGIT cluster (Fig. 2f).

Co-staining the three IRs and counting the individual IR-positive cells from a total of 104,236 cells in COHORT 1 samples revealed that most IR-positive cells expressed only one marker.

In summary, an immunosuppressive microenvironment appears in tumours belonging to the LAG-3 cluster (i.e., ranging from acquired immunity to innate immunity in primary ccRCC), notably revealing high levels of cell exhaustion in tumour-infiltrating CD8 T-cells and infiltrating TAMs.

**Metastasis-specific differences**. We next investigated metastasis-specific differences in the three new IR clusters by applying ccRCC tumour samples harbouring metastases in the lung, bone, viscera, brain, and other sites (i.e. COHORT 2, Supplementary Table 2). Automated single-cell counting was applied, and LAG-3/TIM-3/TIGIT-positive cells were counted, and the mean percentages of receptor-positive cell density were 4.1% for LAG-3, 6.6% for TIM-3, and 2.1% for TIGIT in a total of 47 ccRCC metastases. Co-staining the three IRs and counting the individual IR-positive cells from a total of 103,098 cells in COHORT 2 samples revealed that most IR-positive cells expressed single markers (Supplementary Fig 1d). Together, our results suggest that the cells in metastatic regions can dominantly express one of the three IRs in ccRCC.

Hierarchical clustering of the 47 metastatic ccRCC tumour samples based on the normalized LAG-3-, TIM-3-, and TIGIT-

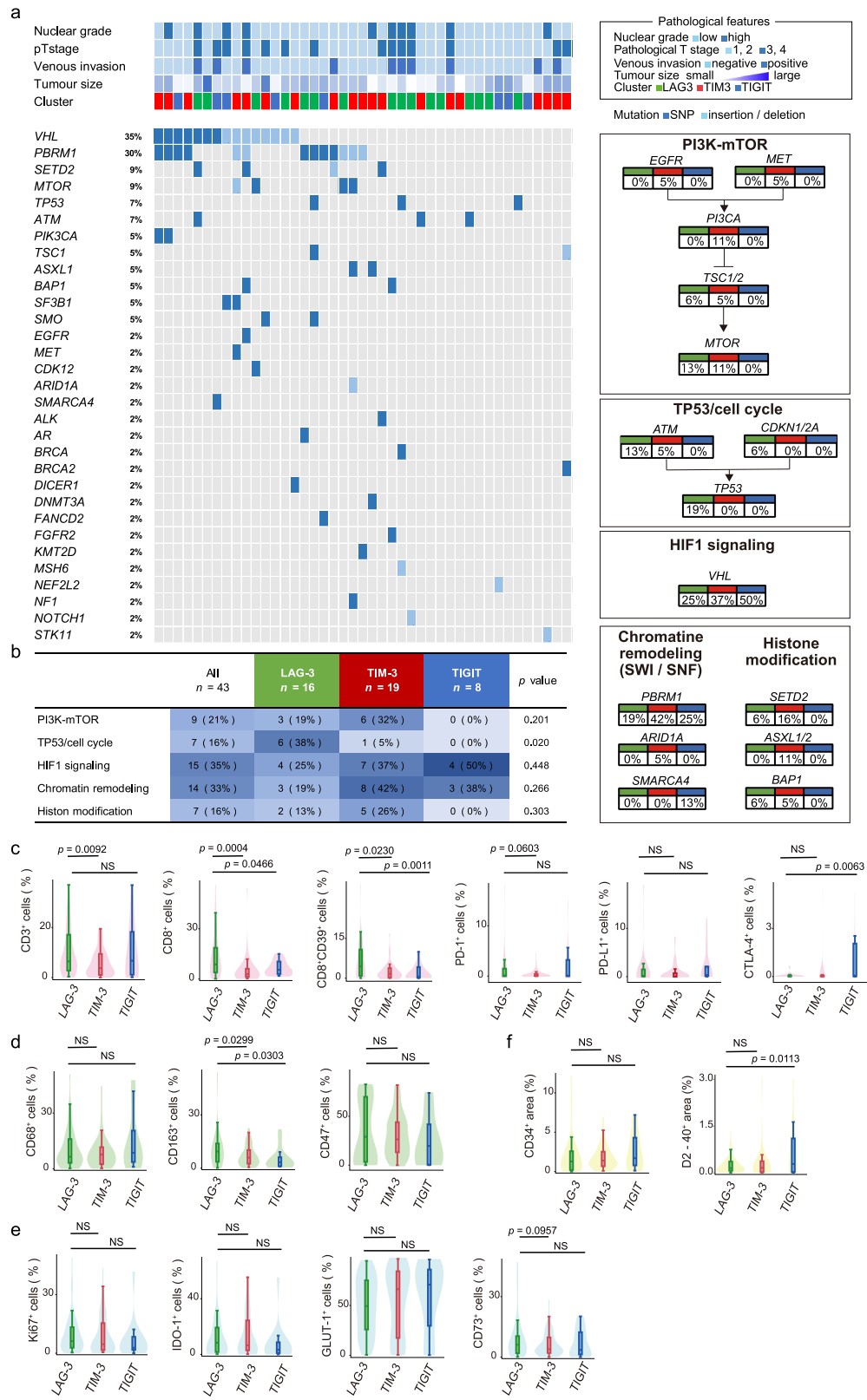

positive cell densities identified three groups with distinct IR levels. TIM-3 (47%), LAG-3 (30%) and TIGIT (23%) were individually dominant in independent clusters (Fig. 3a). Since no significant association was revealed on statistical analysis between the IR profile patterns and the metastatic regions (Supplementary Table 2), regional heterogeneity added another layer of complexity to the second series of IRs (LAG-3, TIM-3, and TIGIT) in

ccRCC. Taken together, these results suggest that lung metastasis has a unique characteristic and show that the TIGIT cluster occupied the majority (Fig. 3b). On the other hand, the TIM-3 cluster occupied the majority in the brain and visceral metastasis (Fig. 3b).

Next, cell-by-cell immunohistological analysis for acquired immunity was performed using the metastatic ccRCC samples.

**Fig. 2 Immunogenomic differences underlying the phenotypic signatures of the new IRs (LAG-3, TIM-3, and TIGIT). a** Alteration landscape of 43 primary ccRCC tumour samples. Upper heatmap: nuclear grade, tumour stage, venous invasion, tumour size and information on the three IR signatures. **b** Genomic alteration differences in tumorigenic signalling pathways related to ccRCC development and the three IR (LAG-3, TIM-3, and TIGIT) signatures. The table shows the percentage of samples with alterations in each of the selected signalling pathways. *p* values were determined with a two-tailed Fisher's exact test. **c–f** Positive cell densities and areas (CD34 and D2-40) in 105 ccRCC samples obtained from patients based on the three new IR clusters and the acquired immunity (**c**), innate immunity (**d**), inhibitory tumour metabolism (**e**), and vascular attribute (**f**) signatures. The lines within the boxes represent the medians, the upper and lower ends of the boxes represent the upper and lower quartiles, and the bars represent the minimum and maximum values in 1.5 times the IQR. *p* values were determined with a two-tailed Mann–Whitney *U*-test. Source data are provided as a Source data file.

The levels of tumour-infiltrating CD3 and CD8 T-cells were higher in metastatic lesions from the LAG-3 cluster than in those from the two other clusters (Fig. 3c). Tumours in the LAG-3 cluster were also associated with high levels of CD39 in CD8 T-cells and PD-L1 (Fig. 3c). However, our immunohistological analysis of innate immunity showed fewer differences in the levels of infiltrating TAMs among the three clusters (Fig. 3d). Our assessments also extended to the state of tumour metabolism in metastatic ccRCC samples labelled with the proliferative Ki-67, IDO-1, GLUT-1, and CD73 markers and revealed that the LAG-3 cluster had significantly higher expression of Ki-67 than in those from the two other clusters and higher expression of IDO-1 than those in the TIGIT cluster (Fig. 3e). The cancer stroma molecules CD34 and D2-40 were not different among the three groups (Fig. 3f).

In summary, the immunosuppressive microenvironment could be continued in tumour metastases in the LAG-3 cluster, and high levels of inhibitory tumour metabolism might be also maintained in cancer cells in the LAG-3 cluster in metastatic ccRCC.

**Histological subtype-specific differences**. We further examined subtype-specific differences in the three IR clusters by applying primary non-ccRCC tumour samples of the papillary, chromophobe, sarcomatoid, Xp11.2 translocation, and collecting duct subtypes (i.e. COHORT 3, Supplementary Table 3) to our platform. Automated single-cell counting was performed, and LAG-3/TIM-3/TIGIT-positive cells were counted, and the mean percentages of receptor-positive cell density were 2.7% for LAG-3, 6.7% for TIM-3, and 3.3% for TIGIT in a total of 41 non-ccRCC individuals. Co-staining the three IRs and counting the individual IR-positive cells from a total of 118,579 cells in the COHORT 3 samples revealed that most IR-positive cells expressed single markers (Supplementary Fig 1e). Together, our results suggest that the cells in the tumour area can dominantly express one of the three IRs in non-ccRCC.

Hierarchical clustering of the 41 non-ccRCC tumour samples based on the positive cell densities for the three IRs identified three groups with distinct IR levels. TIM-3 (49%), LAG-3 (10%) and TIGIT (41%) were individually dominant in independent clusters (Fig. 4a). Interestingly, hierarchical clustering of non-ccRCC tumours revealed that TIGIT cluster was rather majority and prominent in a subset of non-ccRCC (Fig. 4b). However, no statistical association was noted between the IR profile patterns and the non-ccRCC histological subtypes (Supplementary Table 3). On the other hand, variations in the patterns of the three IR signatures across histological subtypes were obvious, indicating that inter-subtype heterogeneity among non-ccRCCs has added another layer of complexity to the second series of IRs (LAG-3, TIM-3, and TIGIT). Taken together, these results show that the chromophobe subtype had slightly different characteristics in non-ccRCCs and the TIM-3 cluster occupied the majority (Fig. 4b). Cell-by-cell immunohistological analyses for acquired immunity, innate immunity, inhibitory tumour metabolism, and

cancer stroma were performed in the same fashion, but no difference was noted in any field (Supplementary Fig 2).

**Validation and discrimination of the LAG-3, TIM-3, and TIGIT signatures**. The profiling of ccRCC patients with regard to the three new IR signatures may result in robust screening when combined with the second series of IRs and anti-PD-1/PD-L1 therapies. However, is the exclusive presence of each IR across individual clusters universal among RCCs? To answer this question, we sought to validate our hypothesis using two publicly available confirmation cohorts: the TCGA (Fig. 5a) and Sato (Fig. 5b) ccRCC datasets[27,29]. Similar results were obtained from the two large-scale RNA-sequencing datasets, revealing that each transcription dataset from the two ccRCC cohorts was successfully distributed into three different clusters with the individual dominations of each IR level. Interestingly, profiling pan-cancer analysis of the TCGA dataset for 14 different solid tumours also revealed distinct three clusters by the levels of these new IRs and were classified into three clusters with distinct IR levels (Supplementary Fig 3), supporting our idea may have a broad impact on other cancer types in medicine.

We then asked how we should judge the three IR signatures in each tumour in clinical practice. In summary, our automated platform for immunohistologically discriminating LAG-3, TIM-3, and TIGIT signatures at the single-cell level may provide a quantitative and high-throughput pathological assessment. We herein propose an optimal workflow and biomarker cut-off for the three IRs to translate in future practice (Fig. 5c). First, COHORT 1 consisted of FFPE tumour samples, which are stored in vast numbers in biobanks worldwide. We referred to AUC-ROC curves (Supplementary Fig 4) to analyse automated single-cell pathology data from COHORT 1 to test our workflows and determine potential biomarker cut-offs for the three IRs. The results revealed that patients in this training group were successfully divided into three groups with distinct IR levels by immunohistochemistry (Fig. 5d). No case in which all three IR expression intensities belong to the percent tiles with less than 5% existed in our model, suggesting either IR expression is predominant in individual ccRCC tumours.

Since tumours are often heterogeneous, we next examined the integrity of IR profiles within the same tumour and investigate how those different parts in tumour area correlated with each other. We applied each IR positive cell density obtained from 5 different regions to the diagnostic model as shown in Fig. 5c. Results revealed that mean concordance rate of 85% for multi-region samplings from a single tumour (Supplementary Fig 5). For clinical use, our model was finally applied to the external validation dataset containing in-house primary ccRCC samples, namely, COHORT 4 (*n* = 96) (stored in alcohol-based fixative; see Methods). Although tumour characteristics in COHORT 4 differ from that in COHORT 1, e.g., lower tumour grade/pT stage and smaller tumour size in COHORT 4 (Table 2), we achieved similarly good discrimination in screening matched IR clusters by immunohistochemistry in COHORT 4 (Fig. 5e).

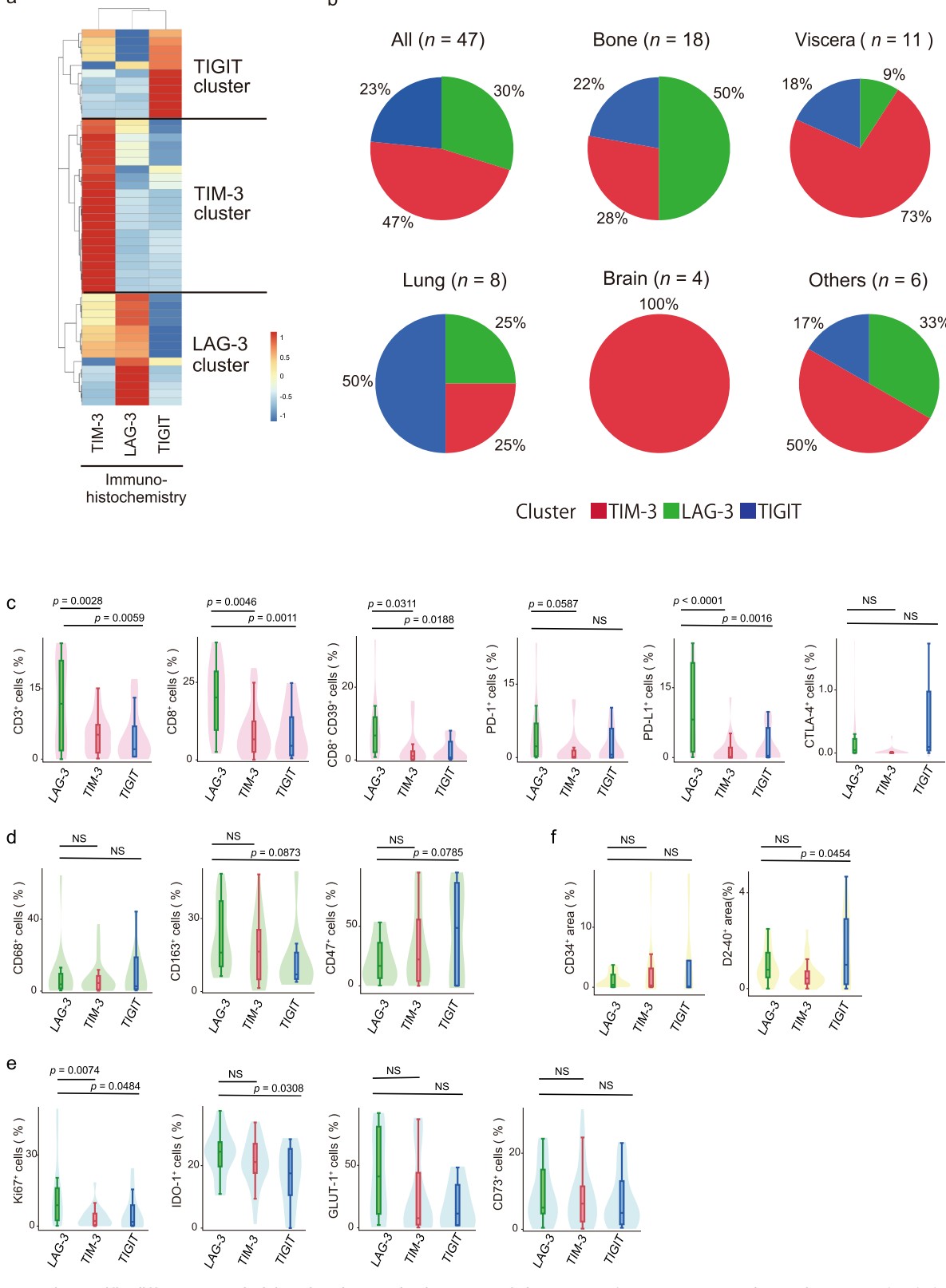

**Fig. 3 Metastasis-specific differences underlying the phenotypic signatures of the new IRs (LAG-3, TIM-3, and TIGIT). a** Hierarchical clustering heatmap (low, blue; high, red) using positive-stained cell densities of LAG-3, TIM-3, and TIGIT in 47 metastatic ccRCC samples. Data were normalized prior to clustering. **b** Inferred IR cluster distributions obtained in (**a**) by metastatic region. **c–f** Positive cell densities and areas (CD34 and D2-40) in 47 metastatic ccRCC samples obtained from patients based on the three new IR clusters and the acquired immunity (**c**), innate immunity (**d**), inhibitory tumour metabolism (**e**), and vascular attribute (**f**) signatures. The lines within the boxes represent the medians, the upper and lower ends of the boxes represent the upper and lower quartiles, and the bars represent the minimum and maximum values in 1.5 times the IQR. *p* values were determined with a two-tailed Mann–Whitney *U*-test. Source data are provided as a Source data file.

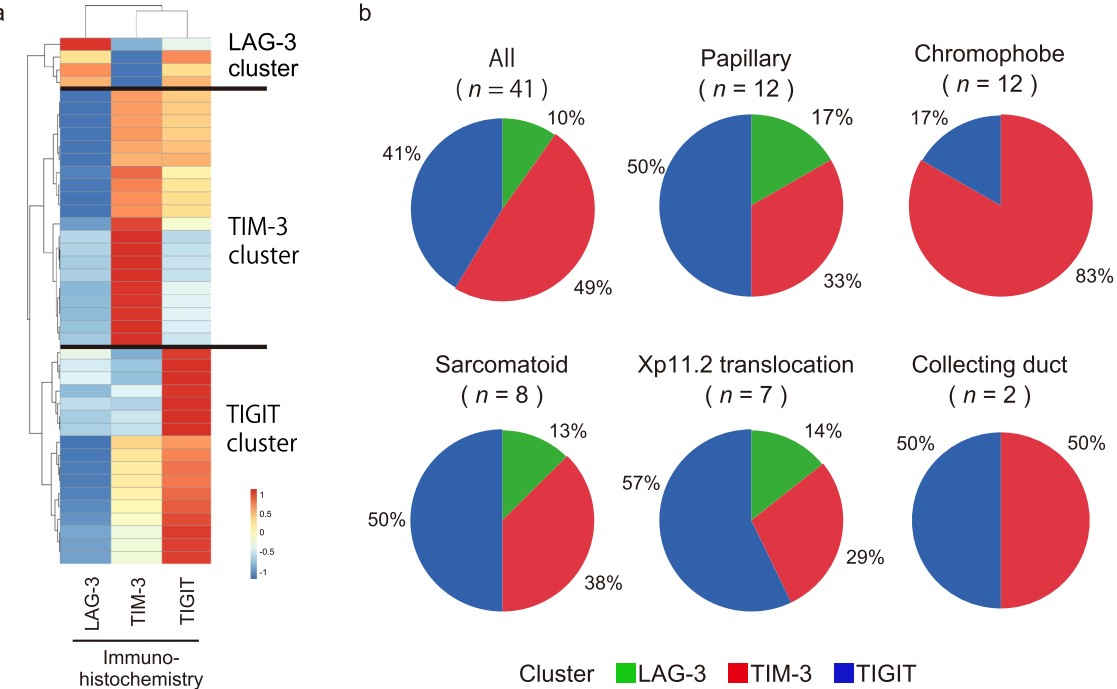

**Fig. 4 Histological subtype-specific differences underlying the phenotypic signatures of the new IRs (LAG-3, TIM-3, and TIGIT). a** Hierarchical clustering heatmap (low, blue; high, red) using positive-stained cell densities of LAG-3, TIM-3, and TIGIT in 41 non-ccRCC samples. Data were normalized prior to clustering. Samples were obtained from the papillary ($n = 12$), chromophobe ($n = 12$), sarcomatoid ($n = 8$), Xp11.2 translocation ($n = 7$), and collecting duct ($n = 2$) subtypes. **b** Inferred IR cluster distributions obtained in (**a**) by histological subtype.

Prognostic analysis of the validation cohort, COHORT 4, showed that the recurrence-free survival (Fig. 5f) and overall survival (Fig. 5g) rates were different among the three IR clusters, in which the LAG-3 cluster was associated with the worst recurrence-free survival and overall survival rate. Furthermore, in multivariable analysis, the LAG-3 cluster ($p = 0.020$, HR: 4.69 with the TIGIT cluster as reference; $p = 0.010$, HR: 4.81 with the TIM-3 cluster as reference) was a significant risk factor for overall mortality following surgery, independent of advanced pathological T stage and positive venous invasion (Supplementary Table 4, Supplementary Data 2).

## Discussion

Despite rapid progress in recent immunotherapies, the overall response rate of existing anti-PD-1/PD-L1 and/or anti-CTLA-4 therapies in patients is far from satisfactory in RCC. In a complicated tumour immune environment, the blockade of a single immune checkpoint molecule may induce complementary changes in other immune modulators[19,20,30–33]. Importantly, therapies targeting new IRs (LAG-3, TIM-3, and TIGIT) are now being applied in clinical trials (registered at ClinicalTrial.gov) or are under active development[20].

A growing body of evidence in preclinical models has also shown potential synergies blocked by combining the first series (i.e. PD-1, PD-L1, and CTLA-4) and second series (i.e. LAG-3, TIM-3, and TIGIT) of IRs[20]. The co-expression of LAG-3 and PD-1 has been observed on intratumoural T-cells in a subset of mouse xenograft tumours and human RCCs[34–36]. Compensatory mechanisms of upregulated TIM-3 have been observed in patients with non-small-cell lung cancer when developing acquired resistance to anti-PD-1 therapy[32]. The combination of TIGIT and PD-1 blockade is involved in CD8$^+$ T-cell function in patients with melanoma[37]. So far, these data may support the use of rational combinations of the three new IRs for an optimal immunotherapy approach. Thus, our next aim should be to

determine how to properly use agents that target the second series of IRs in future practice. Is there any useful biomarker that can be translated to the clinic?

Individual IR levels may be a valid biomarker if robust assessments by immunohistochemistry are allowed in a high-throughput manner[5]. In this study, our immunohistological platform allowed us to show distinct patterns of the three new IRs in human RCC samples, whose expression was exclusively dominant regardless of primary or metastatic disease. This finding indicates that human RCC phenotypes can be divided into three clusters based on the immunohistological levels of the three new IRs, although small percentages of overlap may be present. We further deepened our understanding of the unique characteristics of these IR profiles; for example, the LAG-3 cluster showed a stronger association with immunosuppression than the other two clusters, and this tendency was clearly inherited by metastatic lesions. We also determined an optimal workflow and biomarker cut-off for immunohistologically profiling the three IR levels using internal and external validation cohorts of ccRCC patients. Notably, application of the automated single-cell count system enables rapid and quantitative screening in the setting of clinical pathology.

In this study, we revealed that individual IR levels with exclusive domination in each type of tumour can be the promising tool for profiling human RCC. We suggest that these new immune checkpoints combined with anti-PD-1/PD-L1 treatment may provide further therapeutic effects in progression after immunotherapy, since rechallenges of anti-PD-1/PD-L1 therapies after relapse may be beneficial in a fair percentage of patients[38]. However, some limitations remain to be addressed at this stage. First, our study was retrospective, and a limited number of patients were included in the analysis. Second, little data are yet available to biologically support the significance of IR expression in RCC. Third, ligands belonging to the B7 family (B7-H3, B7-H4, B7-H5) are also attracting attention as a new immuno-

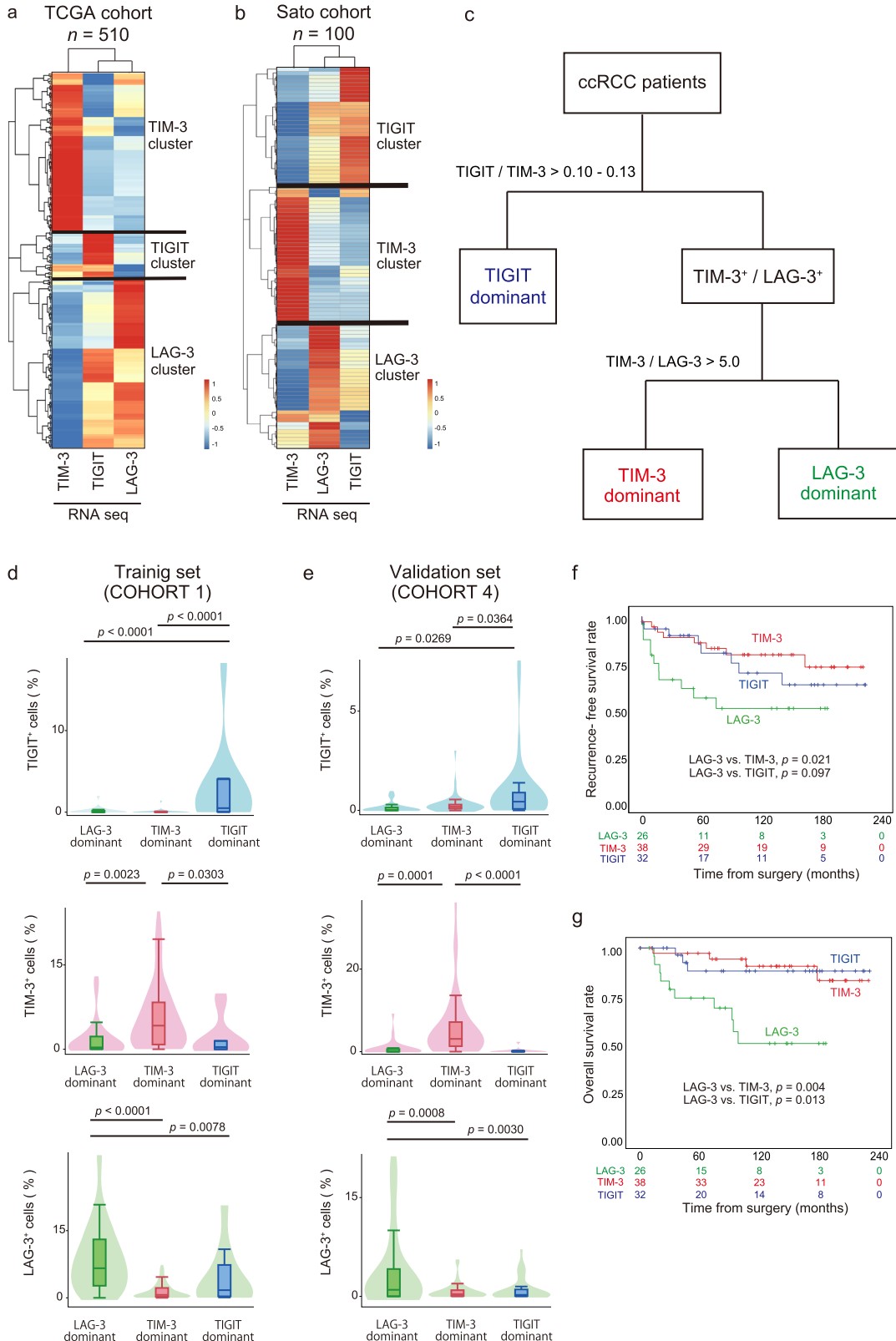

oncology approach[19]. However, there are many gaps in understanding the role of the individual members of this family, and we herein assume that ligand pathway-based approaches such as this represent the next wave in medicine. Last, it is necessary to evaluate whether individual IR levels are useful in predicting the effects of matched anti-IR agents in real-world data. Thus, since the goal of this study is to demonstrate that conclusions such as

this could be drawn using our method, future studies including prospective and/or large-scale dataset validations are needed to provide robust evidence for the proper use of agents targeting the second series of IR agents.

In conclusion, our platform for discriminating the tumour profiles of three IRs, LAG-3, TIM-3, and TIGIT, may have an impact on predicting both outcome and the immune

**Fig. 5 Validation and schematic workflow for clinically discriminating the phenotypic signatures of the new IRs (LAG-3, TIM-3, and TIGIT) by automated single-cell pathology. a, b** Hierarchical clustering heatmap (low, blue; high, red) using inferred IR signatures from RNA-seq data analysis of the TCGA KIRC (**a**) and Sato (**b**) cohorts. **c** Flowchart for discriminating the LAG-3, TIM-3, and TIGIT signatures by immunohistochemistry. **d, e** Inferred IR level distributions of the training cohort (**d**), (COHORT 1, $n = 105$), and the validation cohort (**e**), (COHORT 4, $n = 96$) obtained in (**c**). The lines within the boxes represent the medians, the upper and lower ends of the boxes represent the upper and lower quartiles, and the bars represent the minimum and maximum values in 1.5 times the IQR. $p$ values were determined with a two-tailed Mann–Whitney U-test. Source data are provided as a Source data file. **f, g** Kaplan–Meier survival curves for recurrence-free survival (**f**) and overall survival (**g**) following surgery in 96 ccRCC patients from the validation cohort (COHORT 4) based on the inferred IR signatures from automated single-cell pathology. $p$ values were determined with log-rank test. Source data are provided as a Source data file.

**Table 2 Clinicopathological characteristics associated with the phenotypic signatures of the new IRs (LAG-3, TIM-3, and TIGIT) in a validation group of 96 primary ccRCC tumour samples (COHORT 4).**

| Characteristic | All patients ($n = 96$) | $p$ value COHORT 4 vs COHORT 1 | IR profile | | | $p$ value | | |
| --- | --- | --- | --- | --- | --- | --- | --- | --- |
| | | | LAG-3 $n = 26$ (27%) | TIM-3 $n = 38$ (40%) | TIGIT $n = 32$ (33%) | LAG-3 vs TIM-3 | LAG-3 vs TIGIT | TIM-3 vs TIGIT |
| Age, yr, median, (IQR) | 60 (52–68) | 0.412 | 63 (55–73) | 57 (52–70) | 65 (51–70) | 0.682 | 0.420 | 0.653 |
| Gender, no (%): | | 0.192 | | | | 0.132 | 0.061 | 0.745 |
| Male | 77 (80%) | – | 17 (65%) | 32 (84%) | 28 (88%) | – | – | – |
| Female | 19 (20%) | – | 9 (35%) | 6 (16%) | 4 (13%) | – | – | – |
| Nuclear grade, no (%): | | 0.006 | | | | 0.010 | 0.061 | 0.695 |
| G1 + G2 | 80 (83%) | – | 17 (65%) | 35 (92%) | 28 (88%) | – | – | – |
| G3 + G4 | 16 (17%) | – | 9 (35%) | 3 (8%) | 4 (13%) | – | – | – |
| Pathological T stage, no (%): | | <0.001 | | | | 0.293 | 0.274 | 1.000 |
| pT1 + pT2 | 83 (86%) | – | 20 (77%) | 34 (89%) | 29 (91%) | – | – | – |
| pT3 + pT4 | 13 (14%) | – | 6 (23%) | 4 (11%) | 3 (9%) | – | – | – |
| Venous invasion, no (%): | | 0.523 | | | | 0.132 | 0.563 | 0.381 |
| Yes | 23 (24%) | – | 9 (35%) | 6 (16%) | 8 (25%) | | | |
| No | 73 (76%) | – | 17 (65%) | 32 (84%) | 24 (75%) | | | |
| Tumour size, mm, median (IQR) | 40 (30–52) | <0.001 | 49 (28–60) | 35 (30–50) | 37 (29–50) | 0.461 | 0.269 | 0.666 |

$p$ values were determined with a two-tailed Mann–Whitney U-test or Fisher's exact test.

microenvironment in RCC. Furthermore, this IRs classification may constitute a framework for investigating immunotherapy responses to these IRs in clinical trials.

## Methods

**Human tumour samples**

*Training groups: COHORTS 1–3.* After approval from the Institutional Review Board, formalin-fixed and paraffin-embedded (FFPE) tumour samples obtained from 1999-2017 were randomly collected from three different patient cohorts at Keio University Hospital (Tokyo, Japan) based on histological type, pathological T stage, and systemic therapy. The UICC TNM system was used for tumour staging, and nuclear grading was carried out according to the WHO/International Society of Urological Pathology grading system. The details of the three groups are as follows: COHORT 1, primary ccRCC tumours treated surgically ($n = 105$, Table 1); COHORT 2, ccRCC tumour metastases diagnosed histologically ($n = 47$: lung, 8; bone, 18; viscera, 11, brain, 4, and others, 6; Supplementary Table 2); and COHORT 3, primary non-ccRCC tumours ($n = 41$: papillary, 12; chromophobe, 12; sarcomatoid, 8; Xp11.2 translocation, 7; and collecting duct, 2; Supplementary Table 3). Three-millimetre cores were punched out from the optimal tumour areas of each sample.

*Validation group: COHORT 4.* Samples from COHORT 4, consisting of patients with primary ccRCC tumours treated surgically ($n = 96$, Table 2), were independently collected as an external validation cohort and compared with the training cohort from the same institution. COHORT 4 differs from COHORTS 1–3 in that an alcohol-based PAXgene (Qiagen) was used for fixation prior to paraffin embedding, and 4-mm cores were punched out from the optimal tumour areas of each sample.

No statistical methods were used to predetermine the sample size. All samples were identified with numbers to avoid investigator bias during tissue preparation

and data analysis. No human samples were excluded from the analysis. All procedures were performed in approval of the Research Ethics Committee of Keio University (Approval No-20180098 and 20190059) and in compliance with the 1964 Helsinki Declaration and present ethical standards. The samples were residual from a clinical examination without using any identifiable information of the individuals or the application of any intervention. Participation in the study was optional. Both written informed consent and passive (opt-out) informed consent procedures have been applied to the experimental use of human samples. Opt-out informed consent from patients was obtained by exhibiting the research information on our department website (Department of Urology, Keio University Hospital, Tokyo, Japan). All participant patients or families of deceased patients could withdraw consent by contacting the researcher with a 24-h phone number. The need to obtain written informed consent was waived if patients had finished their follow-up or had died, due to the study's observational nature and the urgent need for cancer patient care. This was approved and reviewed by the Research Ethics Committee of Keio University, in accordance with the ethical guidelines for Medical and Health Research Involving Human Subjects (Public Notice of the Ministry of Education, Culture, Sports, Science and Technology and the Ministry of Health, Labor and Welfare as of July 2018: https://www.lifescience.mext.go.jp/files/pdf/n2181_01.pdf)[39].

**Immunohistochemistry.** Tissue microarray samples were cut into 5-μm–thick sections and placed onto silane-coated glass slides. After deparaffinization, the sections were processed for antigen retrieval and blocking. Thereafter, all sections were incubated overnight with primary antibodies (Supplementary Table 5), followed by incubation with secondary antibodies conjugated to a peroxidase-labelled dextran polymer. Colour development for immunohistochemistry was achieved using 3,3'-diaminobenzidine in 50 mM Tris-HCl (pH 5.5) containing 0.005% hydrogen peroxidase. Finally, sections treated with 3,3'-diaminobenzidine were counterstained with haematoxylin. Multiplex immunohistochemistry was performed with an Opal Multiplex IHC Kit (PerkinElmer, Waltham, Massachusetts,

USA). After deparaffinization, the sections were subjected to antigen retrieval and blocking, incubated with primary antibodies for 4 h and then incubated with Polymer HRP Ms + Rb as the secondary antibody for 10 min at room temperature. Opal fluorophores were pipetted onto each slide for 10 min at room temperature, and the slides were then microwaved to strip the primary and secondary antibodies. We repeated the same protocol using the next primary antibody targets; later, DAPI was pipetted onto each slide for 5 min at room temperature. The slides were covered with ProLong® Diamond Antifade Mountant (Thermo Fisher Scientific). Images were acquired using a fluorescence microscope (IX-73, Olympus). The experiment was performed one time because of using human sample.

**Automated single-cell analysis**. All 3,3'-diaminobenzidine-stained sections were scanned using a high-resolution digital slide scanner (NanoZoomer-XR C12000; Hamamatsu Photonics, Hamamatsu, Shizuoka, Japan). Then, both immunohistochemical signals and nuclei were separately segmented by a computerized image analysis system (HistoQuest software, TissueGnostics, Vienna, Austria). Finally, cells with both overlapping segmented immune signals and nuclei were automatically counted and normalized as the positive cell density obtained to the total cell density from the same sample region harbouring all punched cores. The CD34 (to label blood vessels) and D2-40 (to label lymph vessels) immunosignals in the tumours were quantified in each core as the percentage of positive area after automated signal segmentation. Analyses were reproduced twice. Analysis of cellular images obtained following use of the Opal Multiplex IHC Kit system was performed using inForm software ver. 2.4 (PerkinElmer), allowing the separation and measurement of spectrally overlapping markers in current multiplex assays at the single-cell level. First, cell segmentation was performed by identifying the locations of individual cell nuclei using a counterstain-based approach with DAPI. Multiplexed phenotyping was then performed using an intensity-based threshold that was determined manually in the inForm images. Individual positive cell density data were normalized in the range 0–1, with the bottom 10% as the lower limit and the top 10% as the upper limit, and then used for hierarchical clustering by Ward's linkage methods in R software.

**DNA extraction and sequencing**. Genomic DNA was extracted from fresh-frozen tissue samples from the selected tumour areas with a DNeasy Blood & Tissue Kit (Qiagen) according to the manufacturer's protocol. The DNA integrity number was 4.0, which was calculated using an Agilent 2000 TapeStation (Agilent Technologies, Waldbronn, Germany). A genomic DNA library was constructed using GeneRead DNAseq Targeted Panel V2 (Human Comprehensive Cancer Panel), which covers more than 95% of the total exon region in 160 cancer-related genes[40]. The library was amplified using a GeneRead DNA I Amp Kit (Qiagen) and sequenced using MiSeq (Illumina). The FastQ files obtained from MiSeq (Illumina) were analysed using an original bioinformatics pipeline called GenomeJack (Mitsubishi Space Software, Tokyo, Japan)[40].

**RNA-sequencing dataset analysis**. We analysed the TCGA dataset for kidney renal clear cell carcinoma[27], for which RNA-sequencing data were downloaded from cBioPortal[41]. Additionally, we used the Sato dataset, consisting of RNA-sequencing data from ccRCC patients, which were downloaded from EGA database[29]. Furthermore, a pan-cancer TCGA dataset covering 14 cancer types, i.e. head and neck cancer (HNCA), oesophageal cancer (OECA), lung cancer (LUCA), breast cancer (BRCA), hepatocellular carcinoma (HECA), melanoma (MELA), uterine corpus endometrial carcinoma (CECA), bladder urothelial cancer (BLCA), colorectal cancer (COCA), ovarian cancer (OVCA), prostate cancer (PRCA), glioma (GILO), gastric cancer (GACA), and thyroid cancer (THCA), was also downloaded from cBioPortal[42]. The mRNA expression z-score data for LAG-3, TIM-3 and TIGIT were used for hierarchical clustering in the same manner as described above.

**Statistics**. Values are presented as the means with standard errors or medians with interquartile ranges (IQR) for continuous variables and frequencies with percentages for categorical variables. Variables between groups were compared using Fisher's exact test and the Mann–Whitney U-test, as appropriate. We performed receiver operating characteristic (ROC) curve analysis using the data from the training group (COHORT 1) to define a potential cut-off cell density that discriminates the phenotypic signatures of the LAG-3, TIM-3, and TIGIT in ccRCCs. An area under the curve (AUC) value of 1.0 represents perfect discrimination, and a value of 0.5 represents no discrimination. Univariable and multivariable Cox regression models with stepwise selection were used to evaluate variables associated with recurrence-free and overall survival. Survival curves were estimated using the Kaplan–Meier method and compared using the log-rank test. The hclust R package (version 3.6.1) was used for hierarchical clustering of tumour samples. Statistical significance was accepted for $p < 0.05$. All analyses were performed using R statistical language version 3.6.1 (R Foundation for Statistical Computing, Vienna, Austria), SPSS version 24.0 (IBM-SPSS Inc., Tokyo, Japan) and JMP version 15.0 (SAS Institute Inc., Cary, NC, USA).

**Reporting summary**. Further information on research design is available in the Nature Research Reporting Summary linked to this article.

## Data availability

The DNA-seq data have been deposited in the SRA database under accession code PRJNA 739032. The TCGA dataset for kidney renal clear cell carcinoma in Fig. 5 was downloaded from the cBioPortal (https://www.cbioportal.org/study/summary?id=kirc_tcga_pan_can_atlas_2018; access date: April, 2020)[27]. The SATO dataset in Fig. 5 was downloaded from the EGA database (https://ega-archive.org/datasets/EGAD00001000597; access date: April, 2020)[29]. The TCGA datasets from 14 cancer types in Supplementary Fig 3 were downloaded from the cBioPortal (https://www.cbioportal.org/study/summary?id=hnsc_tcga_pan_can_atlas_2018; https://www.cbioportal.org/study/summary?id=esca_tcga_pan_can_atlas_2018; https://www.cbioportal.org/study/summary?id=lusc_tcga_pan_can_atlas_2018%2Cluad_tcga_pan_can_atlas_2018; https://www.cbioportal.org/study/summary?id=brca_tcga_pan_can_atlas_2018; https://www.cbioportal.org/study/summary?id=lihc_tcga_pan_can_atlas_2018; https://www.cbioportal.org/study/summary?id=skcm_tcga_pan_can_atlas_2018; https://www.cbioportal.org/study/summary?id=ucec_tcga_pan_can_atlas_2018; https://www.cbioportal.org/study/summary?id=blca_tcga_pan_can_atlas_2018; https://www.cbioportal.org/study/summary?id=coadread_tcga_pan_can_atlas_2018; https://www.cbioportal.org/study/summary?id=ov_tcga_pan_can_atlas_2018; https://www.cbioportal.org/study/summary?id=prad_tcga_pan_can_atlas_2018; https://www.cbioportal.org/study/summary?id=gbm_tcga_pan_can_atlas_2018; https://www.cbioportal.org/study/summary?id=stad_tcga_pan_can_atlas_2018; https://www.cbioportal.org/study/summary?id=thca_tcga_pan_can_atlas_2018; access date April, 2021)[41]. The source data underlying all Figures and Supplementary Figures are provided as a Source Data file. All the other data supporting the findings of this study are available within the article and its Supplementary Information files. Due to privacy concerns restricted by the ethics, clinical data of individual patients (i.e., indirect identifiers patient age and gender in Supplementary Data 1–2) will be shared, after proper de-identification, upon reasonable request to the corresponding author from colleagues who want to analyse in deep our findings. Also, remaining data will be shared upon reasonable request to the corresponding author. Source data are provided with this paper.

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

## Acknowledgements

This study was supported by Grants-in-Aid for Scientific Research (KAKENHI 19K16807 to K.T.; 18H04906, 18K19482, and 19H03792 to N.T.; 18K09150 to T.S.; and 18H02939 to M.O.), the Takeda Science Foundation (N.T.), the Kobayashi Foundation for Cancer Research (N.T.), the SGH Cancer Research Grant (N.T.) and the Keio Gijuku Academic Development Funds (N.T.). The authors thank JKiC (JSR-Keio University Medical and Chemical Innovation Center) for special assistance to the multiplexed fluorescence imaging system. Also, the authors thank the Fourth Laboratory of the Department of Pathology in Keio University School of Medicine for immunohisto-chemical analyses.

## Author contributions

N.T. and K.T. designed the experiments. K.T. performed and analysed all the experiments with the help of K.H., R.T., Y.T., T.M., R.K., N.N., S.M. and T. Shinojima. Y.S., H.K. and S.O. provided advice for RNA-seq. data analysis. T. Sasaki, K.K., T.K., F.M. and T.T. provided advise for bioinformatics analysis. E.A., H.N., K.S. and T.I. performed genetic experiments. N.T. and K.T. wrote the manuscript with inputs from R.M. and M.O.

## Competing interests

The authors declare no competing interests.
