## [Peer Review File · Nature Communications]

Profiling the inhibitory receptors LAG-3, TIM-3, and TIGIT in renal cell carcinoma reveals malignancyReviewers' Comments:

Reviewer #1:

Remarks to the Author:

The main topic of the manuscript is really interesting, but there are notable shortcomings:

1) The Introduction section is poor and messy. It seems that immune checkpoints have only been approved for renal cancer; only in a sentence, the authors mention their use in other tumors. This section needs to be completely revised.

a. First an introduction is needed regarding the role of immune checkpoints in modern oncology; only later, the authors can focus their attention on kidney cancer

b. Page 2 line 15-20. In kidney cancer, Nivolumab and Ipilimumab are just some of the immune checkpoint inhibitors that have demonstrated efficacy: Pembrolizumab and Avelumab have also demonstrated their efficacy, both in combination with Axitinib. Why don't the authors mention these drugs too?

c. Page 2 line 25. The authors focus on LAG-3, TIM-3 and TIGIT, but it should be noted that there are other immune checkpoints, of which we still know little

2) The authors evaluated the cases on the basis of the 3 new immune checkpoints, but never mention the 3 prognostic groups (good, intermediate, poor), which showed notable differences also in terms of response to immunotherapy: in fact, the group a good prognosis appears to demonstrate greater benefit with TKIs, unlike the other two groups, which appear to benefit from immune checkpoint inhibitors. It is essential to mention these differences in this context.

3) The authors mention LAG-3, Tim-3 and TIGIT, without ever describing them. It would be useful to mention the peculiar characteristics of each of them.

4) In the Discussion section, the authors explain that the new immune checkpoints may be useful in case of progression after immunotherapy. Given that, as Escudier pointed out in 2017, a fair percentage of patients can still benefit from Nivolumab after progression, and that some data regarding rechallenge after progression have been observed with the combination Nivolumab + Ipilimumab (ASCO 2019), a comment on this would be helpful

5) The article is full of acronyms that are never explained (e.g. LAG-3, IDO-1 etc.). Please enter the explanation of each individual acronym.

Reviewer #2:

Remarks to the Author:

This study uses single cell quantitative immunohistochemistry (IHC) to analyze the expression of three immune inhibitory receptors in RCC samples. The strength of the work is the sheer amount of data they acquired from human tumor samples comprising primary ccRCC, metastatic ccRCC and primary non-ccRCC tumors. They were also able to show that it appears their assays give consistent results and these findings will be of great interest to anyone studying the immune microenvironment of RCCs. However, there are major flaws in some analyses and in the conclusions regarding the value of their prognostic model (which suggests that LAG-3 dominant tumors have worse overall survival).

Issues that need to be addressed:

1. Sample acquisition. Though it is not totally clear from the methods, their data are from microarrays. It is not stated how many samples from each core were assessed for any particular antibody staining so I have to assume they only used a single microarray slice per antibody per tumor. Given that tumors are often extremely heterogeneous, a very important control would have been to take multiple samples from different parts of a single tumor and see how well those parts correlated with each other. Without that information, it is very hard to know the meaning of these data. This

needs to be a figure in any subsequent submission or at least an explanation of why that control is unnecessary must be there.

2. Cohort 1 survival data: 2 Kaplan-Meier plots are shown in figure 1 demonstrating that the LAG-3 dominant cohort had worse RFS and OS. However, there are known risk factors for recurrence and OS already including tumor grade, T stage and tumor size. A cursory look at Table 1 demonstrates that far higher percentages for LAG-3 dominant tumors were higher grade and higher T stage, which probably accounts for much if not all of the worse outcomes for LAG-3 dominant tumors. They did not look at tumor size (though they could probably easily acquire those data if they have tissue and survival data). In order to show that the LAG-3 dominant expression profile has any real significance for RFS and OS, they need to use a multivariable model (or at least stratify the data) taking into account tumor stage, grade (and ideally size). For cohort 4, they do not provide an equivalent table but it needs to be added and the data need to be analyzed in the same way.

3. Relevance of LAG-3 dominant OS differences: if we accept that LAG-3 dominant tumors have worse survival independent of other factors (which the authors have not proven), the value of this finding is far less at the moment than what is concluded by the authors. They state: "In summary, our platform for discriminating the tumour profiles of the IRs LAG-3, TIM-3, and TIGIT may have a broad impact on treatment decision-making and patient counselling in RCC and, more broadly, all cancer types. Furthermore, this new classification may constitute a novel framework to investigate immunotherapy responses to these IRs in clinical trials." The first sentence is incorrect. At the moment, the finding has no impact on decision-making and minimal, if any, impact on patient counseling. This is a prognostic, not a predictive model – thus it has no impact on treatment decisions. The differences in outcome are not so great that one could tell a patient anything about what it would mean for his or her particular tumor. Furthermore, to claim that this has any meaning for other tumor types is without evidence. I agree, however, that it could be used during clinical trials potentially (if it were a valid finding) to assess whether it had impact on any treatments.

4. They claim that LAG-3 dominant tumors have a more immunosuppressed tumor microenvironment. Their evidence is a higher number of CD8+, CD3+ cells and a higher number of CD39+ and CD163+ cells. They claim that CD39 positivity is direct evidence of increased exhaustion in the immune infiltrate. While there are data that CD8+ CD39+ cells are exhausted, the authors provide no evidence that the CD39+ cells are CD8+. In fact, Tregs express CD39+ as well as other cells. Furthermore, there are no functional assays of exhaustion and they only use a single marker, which is very weak evidence. By the same token they say that there are more TAMs based on the CD163+ cells but there was no difference in CD68 levels and that is another commonly used marker for TAMs. In the absence of any functional data or more markers, these findings are inadequate to make those conclusions.

5. They claim there was a 'tendency' toward resistance with TIGIT and LAG-3 to nivolumab therapy based on a waterfall plot (figure 1j). Given the tiny number of samples (3 tigit, eg) no conclusions can be drawn.

6. It would be nice to know if any of their samples don't fit into any of their groups (ie are there samples that are not TIGIT, LAG-3 or TIM-3 dominant?).

7. In general, because they have no functional data (obviously they cannot given that the samples are FFPE), the differences in levels of specific markers, while statistically significant, are not necessarily biologically significant.

8. To me the most interesting finding is that there appears to be mutual exclusivity between expression of each of the 3 IR markers they look at in ccRCC primary tumors. I do not think this has been demonstrated across the board in other cells types. Notably, those relationships seem to be far weaker in the metastatic ccRCC and the non-ccRCC samples (see supplementary figure). However, the statistic they use to analyze the inverse relationships is Spearman rank but they should probably be using a statistic that is better for analyzing inverse relationships since the important finding is not that there is no correlation between the different IR-dominance profiles but rather that they are mutually exclusive.

9. Regarding the differences in IR dominance at different metastatic sites, there are too few samples to conclude anything, even about lung. The most striking finding is the fact that all brain metastases were TIM-3 dominant. These are hypothesis generating findings but no conclusions can be drawn. It would be important to see if matching the primary with met could be done in any of these cases to see

if the IR dominance matches. That would be very valuable.

Reviewer #3:

Remarks to the Author:

Takamatsu et al. report an immunohistochemistry (IHC)-based analysis platform to distinguish subtypes of human renal cell carcinoma (RCC) samples and predict good responders to a new series of inhibitory receptors (IRs), namely, LAG-3, TIM-3, and TIGIT. In the analysis platform, the authors quantified the densities of cells that dominantly express LAG-3, TIM-3, and TIGIT, respectively, in the IHC images obtained for each RCC sample. They found the individual RCC samples could be divided into three subtypes/clusters based on the expression levels of LAG-3, TIM-3, and TIGIT, with each subtype/cluster containing the dominant cell population positive for one of the three IRs. The authors also tried to characterize some properties of LAG-3, TIM-3, and TIGIT subtypes, such as the differential expression levels of some known cancer-associated genes and the differential metastases outcomes among these three subtypes. The results presented in this study could be potentially important, however, the conclusions in this study are not fully supported by their data.

specific comments:

- 1) In Fig.1, the immunostaining of LAG-3, TIM-3, and TIGIT in the IHC images seems to contain a high level of non-specific staining signals.
- 2) The authors argue that the RCC samples can be divided into three subtypes, and each subtype can dominantly contain a cell population that express one of the three IRs. This would imply that the cells in the same RCC sample can dominantly express only one of the three IRs. As a further confirmation, the authors should use multi-color IHC to co-stain the three IRs in the same RCC sample, for example, using sequential IHC imaging as shown in Tsujikawa et al. (Cell Rep. 2017, 19(1): 203–217).
- 3) As the total cell density can vary a lot from region to region, the authors should normalize each receptor-positive cell density obtained to the total cell density from the exact same sample region, and then use the normalized LAG-3, TIM-3, and TIGIT positive cell densities to determine the RCC subtypes.
- 4) In this study, an important implication to use these three IRs as biomarkers to classify RCC tumors into three subtypes is to show that the certain subtype has certain statistically meaningful properties. For example, the author claims that LAG-3 dominant tumors have the lowest survival rates. However, the small sample sizes used in this study and poor statistics presented do not convincingly support these claims.

REBUTTAL LETTER # NCOMMS-20-31986 “Profiling the inhibitory receptors LAG-3, TIM-3, and TIGIT in renal cell carcinoma reveals malignancy”.

Point-by-point answers to the reviewers:

Reviewer #1, expert in RCC subtypes (Remarks to the Author):

1) The Introduction section is poor and messy. It seems that immune checkpoints have only been approved for renal cancer; only in a sentence, the authors mention their use in other tumors. This section needs to be completely revised.

a. First an introduction is needed regarding the role of immune checkpoints in modern oncology; only later, the authors can focus their attention on kidney cancer.

RESPONSE: We appreciate this suggestion and have modified the INTRODUCTION section in the revised manuscript accordingly. To avoid any misunderstanding that immune checkpoint inhibitors are approved only for kidney cancer, we first show the landscapes and recent advances in immuno-oncology in medicine and later continue with the topic of RCC (see lines 46-48).

b. Page 2 see lines 15-20. In kidney cancer, Nivolumab and Ipilimumab are just some of the immune checkpoint inhibitors that have demonstrated efficacy: Pembrolizumab and Avelumab have also demonstrated their efficacy, both in combination with Axitinib. Why don't the authors mention these drugs too?

RESPONSE: In the revised manuscript, we have added a description of the combination of pembrolizumab and avelumab with axitinib in the INTRODUCTION section (see lines 51).

c. Page 2 line 25. The authors focus on LAG-3, TIM-3 and TIGIT, but it should be noted that there are other immune checkpoints, of which we still know little.

RESPONSE: In addition to clinically targeting the second-generation IRs LAG-3, TIM-3 and TIGIT, as seen in current trials, ligands that belong to the B7 family (B7-H3, B7-H4 and B7-H5) are a major focus of future immunotherapeutic approaches [Andrews LP, et al. *Nat Immunol.* 2019; 20: 1425-34]. However, there are many gaps in the understanding of the role of each member of the B7 family; even their receptors have yet to be determined. Thus, we only mention the B7 family as part of the next wave for ligand pathway-based approaches in the DISCUSSION section in the revised manuscript (see lines 300-303).

2) The authors evaluated the cases on the basis of the 3 new immune checkpoints, but never mention the 3 prognostic groups (good, intermediate, poor), which showed notable differences also in terms of response to immunotherapy: in fact, the group a good prognosis appears to demonstrate greater benefit with TKIs, unlike the other two groups, which appear to benefit from immune checkpoint inhibitors. It is essential to mention these differences in this context.

RESPONSE: Similar criticisms have been received from Reviewer #2, who is an expert in immunotherapies/biomarkers. First, we decided to remove the original Fig 1j, i.e., the response to nivolumab in 14 ccRCC patients, from the revised manuscript, as suggested by the reviewers. Instead, to determine the impact of the tumour profile according to the three new IRs on prognosis and predict the response to immunotherapy, we examined the relationship with the IMDC risk criteria in the current guidelines for metastatic RCC. In total, 57 patients demonstrated recurrence and were assigned to the appropriate risk category by the IMDC indicators (favourable: 35%, intermediate: 47%, and poor: 18%). The relationship between IR profiles and the IMDC risk criteria of the 57 patients is shown in the revised Figure 1i. In short, patients in the LAG-3 cluster may have a worse IMDC risk at disease relapse, indicating the validity of the worse overall survival in the LAG-3 cluster, with resistance to existing anti-angiogenic treatment and immunotherapy. We have added these results to the revised manuscript (see lines 121-131).

3) The authors mention LAG-3, Tim-3 and TIGIT, without ever describing them. It would be useful to mention the peculiar characteristics of each of them.

RESPONSE: In the revised manuscript, we have added a summary of the characteristics of each IR in the INTRODUCTION section (see lines 58-64).

4) In the Discussion section, the authors explain that the new immune checkpoints may be useful in case of progression after immunotherapy. Given that, as Escudier pointed out in 2017, a fair percentage of patients can still benefit from Nivolumab after progression, and that some data regarding rechallenge after progression have been observed with the combination Nivolumab + Ipilimumab (ASCO 2019), a comment on this would be helpful.

RESPONSE: This reviewer's comments are interesting when considering the better treatment options for solid tumours. This study was a phenotypic analysis of three IRs based on tumour samples that had not been subjected to immunotherapy (i.e., immunotherapy-naïve samples). In the future, the way in which immunotherapy will change the expression levels of these new IRs in tumours should be discussed, possibly shedding light on the optimization of salvage immunotherapy after relapsing following initial immunology treatment. We have added a new reference by Gul A et al. (J Clin Oncol 2020; 38: 3088-3094), as the reviewer suggested, and further expanded the DISCUSSION section in the revised manuscript (see lines 295-297).

5) The article is full of acronyms that are never explained (e.g. LAG-3, IDO-1 etc.). Please enter the explanation of each individual acronym.

RESPONSE: Thank you for noting these errors. We have included an explanation for all acronyms in the revised manuscript (see lines 58-59 and 157).

Reviewer #2, expert in immunotherapies/biomarkers (Remarks to the Author):

1) Sample acquisition. Though it is not totally clear from the methods, their data are from microarrays. It is not stated how many samples from each core were assessed for any particular antibody staining so I have to assume they only used a single microarray slice per antibody per tumor. Given that tumors are often extremely heterogeneous, a very important control would have been to take multiple samples from different parts of a single tumor and see how well those parts correlated with each other. Without that information, it is very hard to know the meaning of these data. This needs to be a figure in any subsequent submission or at least an explanation of why that control is unnecessary must be there.

RESPONSE: In the construction of the tissue microarray sections, representative 3-mm cores of WHO/ISUP grade tumour centres were utilized. Thus, we have followed the reviewer's recommendation and conducted a total of five new multi-region samplings from single tumours from 30 patients and verified the consistency of the IR profile within the same tumour (see revised Supplementary Figure 5). Applying the IR-positive cell density obtained from each of the 5 regions, we examined the tumour profile of each region for the 30 patients. In our diagnostic algorithm, as shown in the revised Figure 5c, we ultimately revealed a mean concordance rate of 85% for the multi-region samplings. We have added these results to the revised manuscript (see lines 249-253).

2) Cohort 1 survival data: 2 Kaplan-Meier plots are shown in figure 1 demonstrating that the LAG-3 dominant cohort had worse RFS and OS. However, there are known risk factors for recurrence and OS already including tumor grade, T stage and tumor size. A cursory look at Table 1 demonstrates that far higher percentages for LAG-3 dominant tumors were higher grade and higher T stage, which probably accounts for much if not all of the worse outcomes for LAG-3 dominant tumors. They did not look at tumor size (though they could probably easily acquire those data if they have tissue and survival data). In order to show that the LAG-3 dominant expression profile has any real significance for RFS and OS, they need to use a multivariable model (or at least stratify the data) taking into account tumor stage, grade (and ideally size). For cohort 4, they do not provide an equivalent table but it needs to be added and the data need to be analyzed in the same way.

RESPONSE: We have followed the reviewer's recommendation and conducted uni- and multivariable model analyses for overall and recurrence-free survival using COHORTs 1 and 4, in which tumour size was included as a new variable. In COHORT 1, multivariable analysis did not reveal any association between the IR profiles and recurrence-free survival; however, the IR profile was significantly associated with overall survival ($p = 0.037$), independent of patient age, nuclear grade, and tumour size (see revised Supplementary Table 1). Additionally, similar results were obtained in multivariable models for COHORT 4, revealing that the IR profile by three IRs constituted a significant risk factor for overall survival ($p = 0.009$), independent of pathological T stage and positive venous invasion (see revised Supplementary Table 4). In addition to the equivalent table for COHORT 4 (see revised Table 2), we have added these results to the RESULTS section in the revised manuscript (see lines 112-119 and 261-264).

3) Relevance of LAG-3 dominant OS differences: if we accept that LAG-3 dominant tumors have worse survival independent of other factors (which the authors have not proven), the value of this finding is far less at the moment than what is concluded by the authors. They state: “In summary, broad impact on treatment decision-making and patient counselling in RCC and, more broadly, all cancer types. Furthermore, this new classification may constitute a novel framework to investigate immunotherapy responses to these IRs in clinical trials.” The first sentence is incorrect. At the moment, the finding has no impact on decision-making and minimal, if any, impact on patient counseling. This is a prognostic, not a predictive model – thus it has no impact on treatment decisions. The differences in outcome are not so great that one could tell a patient anything about what it would mean for his or her particular tumor. Furthermore, to claim that this has any meaning for other tumor types is without evidence. I agree, however, that it could be used during clinical trials potentially (if it were a valid finding) to assess whether it had impact on any treatments.

RESPONSE: We agree with the reviewer’s comments and have softened our claim and avoided making any definitive conclusions throughout the manuscript as much as possible. Further, we thank the reviewer for suggesting that we focus on this phenomenon in other cancer types. In addition to the analysis of ccRCC data from the TCGA and Sato datasets, we have added pan-cancer analysis using the TCGA dataset for 14 different solid tumours, in which all cancer types were profiled according to the levels of three IRs, resulting in three clusters with distinct IR levels (see Supplementary Figure 3). We have included these results in the RESULTS section in the revised manuscript (see lines 232-235).

4) They claim that LAG-3 dominant tumors have a more immunosuppressed tumor microenvironment. Their evidence is a higher number of CD8⁺, CD3⁺ cells and a higher number of CD39⁺ and CD163⁺ cells. They claim that CD39 positivity is direct evidence of increased exhaustion in the immune infiltrate. While there are data that CD8⁺CD39⁺ cells are exhausted, the authors provide no evidence that the CD39⁺ cells are CD8⁺. In fact, Tregs express CD39⁺ as well as other cells. Furthermore, there are no functional assays of exhaustion and they only use a single marker, which is very weak evidence. By the same token they say that there are more TAMs based on the CD163⁺ cells but there was no difference in CD68 levels and that is another commonly used marker for TAMs. In the absence of any functional data or more markers, these findings are inadequate to make those conclusions.

RESPONSE: We agree with the reviewer’s comments. First, we performed multiplexed immunohistochemistry to validate the role of CD39 as a marker for T-cell exhaustion. Co-staining of CD8 and CD39, however, revealed that double-positive cells accounted for only 12.2% (95% CI: 9.5-14.5%) of all CD39-positive cells among the COHORT 1 patients. Thus, we next investigated the relationship between double-positive (CD8⁺CD39⁺) cells and IR profiles by applying these multiplexed immunohistochemistry data. The results revealed that tumours in the LAG-3 cluster contained significantly more CD8⁺CD39⁺ cells, thus, we replaced the figures in the revised manuscript (see revised Figures 2 and 3 and Supplementary Figure 2) and corrected the manuscript accordingly. Similarly, CD163 was co-stained with CD68 (labels the general population of macrophages, i.e., M1 plus M2) to verify whether the former is an appropriate marker for detecting TAMs in RCC. The results revealed that CD68⁺CD163⁺ cells accounted for 84.5% (95% CI 71.8-97.2%) of all CD163-positive cells, indicating that most CD163 is expressed on macrophages in ccRCC. We also added this result to the revised manuscript (see lines 148-149 and 154-155).

5) They claim there was a 'tendency' toward resistance with TIGIT and LAG-3 to nivolumab therapy based on a waterfall plot (figure 1j). Given the tiny number of samples (3 tigit, eg) no conclusions can be drawn.

RESPONSE: Similar criticisms have been received from Reviewer #1, who is an expert in RCC subtypes. First, we decided to remove the original Fig 1j, i.e., the response to nivolumab in 14 ccRCC patients, from the revised manuscript, as suggested by the reviewers. Instead, to determine the impact of the tumour profile according to the three new IRs on prognosis and predict the response to immunotherapy, we examined the relationship with the IMDC risk criteria in the current guidelines for metastatic RCC. In total, 57 patients demonstrated recurrence and were assigned to the appropriate risk category by the IMDC indicators (favourable: 35%, intermediate: 47%, and poor: 18%). The relationship between IR profiles and the IMDC risk criteria of the 57 patients is shown in the revised Figure 1i. In short, patients in the LAG-3 cluster may have a worse IMDC risk at disease relapse, indicating the validity of the worse overall survival we found in the LAG-3 cluster, with resistance to existing anti-angiogenic treatment and immunotherapy. We have added these results to the revised manuscript (see lines 121-131).

6) It would be nice to know if any of their samples don't fit into any of their groups (ie are there samples that are not TIGIT, LAG-3 or TIM-3 dominant?).

RESPONSE: We have followed the reviewer's recommendations. First, we evaluated the intensity of the expression of each IR, i.e., the normalized positive cell densities, for all patients, and examined the resulting distributions. We found that there was no case in which all three IR expression intensities were within the 5th percentile. In summary, IR expression is predominant in all cases in this population. We have added this information to the RESULTS section in the revised manuscript (see lines 245-247).

7) In general, because they have no functional data (obviously they cannot given that the samples are FFPE), the differences in levels of specific markers, while statistically significant, are not necessarily biologically significant.

RESPONSE: Thank you for noting this point. We agree with the reviewer's comments. Future studies are needed to demonstrate biological significance in this issue, and we added this point to the DISCUSSION section as a limitation for the entire study (see lines 299-300).

8) To me the most interesting finding is that there appears to be mutual exclusivity between expression of each of the 3 IR markers they look at in ccRCC primary tumors. I do not think this has been demonstrated across the board in other cells types. Notably, those relationships seem to be far weaker in the metastatic ccRCC and the non-ccRCC samples (see supplementary figure). However, the statistic they use to analyze the inverse relationships is Spearman rank but they should probably be using a statistic that is better for analyzing inverse relationships since the important finding is not that there is no correlation between the different IR-dominance profiles but rather that they are mutually exclusive.

RESPONSE: Similar criticisms have been received from Reviewer #3, who is an expert in immunohistochemical analysis. According to his/her comments, our results seem to imply that the cells in

the same RCC sample can dominantly express only one of the three IRs. We think that the answer to this question matches the answer to your question here. First, we removed the original Supplementary Figure 1. Second, we performed multi-colour immunohistochemistry with the Opal-Vectra multiplex immunofluorescence system (Perkin Elmer, Waltham, Massachusetts, USA). In COHORT 1, multi-colour immunohistochemistry for the three IRs highlighted the exclusivity of each marker in the cells in the RCC samples. Co-staining of the three IRs revealed that most of the IR-positive cells expressed single markers (see revised Supplementary Figure 1a-c). Since there was only a small degree of overlap, our data may indicate the validity of the current results obtained from the clustering analyses, as shown in revised Figure 1e. Third, we repeated the same assay using COHORTs 2 and 3 (see revised Supplementary Figure 1d-e) and have added all of the data in the RESULTS section in the revised manuscript (see lines 102-107, 172-175 and 207-210).

9) Regarding the differences in IR dominance at different metastatic sites, there are too few samples to conclude anything, even about lung. The most striking finding is the fact that all brain metastases were TIM-3 dominant. These are hypothesis generating findings but no conclusions can be drawn. It would be important to see if matching the primary with met could be done in any of these cases to see if the IR dominance matches. That would be very valuable.

RESPONSE: We agree with the reviewer's comments. The number of metastasis specimens examined in this study was limited, since the goal of this experiment was to demonstrate that conclusions such as this could be drawn using our method. Together, our data highlighted some findings; for example, all brain metastases were TIM-3 dominant according to their three-IR profiles. Therefore, we softened our claim and avoided making a conclusive statement for this analysis as much as possible; rather, we add a limitation that needs additional consideration in future research (see lines 303-307). Further, following the reviewer's recommendation, we studied 7 cases of bone metastases that could be matched to the primary lesion, 4 of which (57%) had the same tumour profile according to the three IRs. However, the number of samples examined here was limited, and no definitive conclusion may be drawn at this stage. Thus, we decided to not include these results in the revised manuscript.

Reviewer #3, expert in immunohistochemical analysis (Remarks to the Author):

1) In Fig.1, the immunostaining of LAG-3, TIM-3, and TIGIT in the IHC images seems to contain a high level of non-specific staining signals.

RESPONSE: Thank you for noting this spot. In this revision, all sections were re-reviewed by a board-certified genitourinary pathologist (S.M.), and we changed all the IR immunohistochemistry images to representative images (see revised Fig 1a-c).

2) The authors argue that the RCC samples can be divided into three subtypes, and each subtype can dominantly contain a cell population that express one of the three IRs. This would imply that the cells in the same RCC sample can dominantly express only one of the three IRs. As a further confirmation, the authors should use multi-color IHC to co-stain the three IRs in the same RCC sample, for example, using sequential IHC imaging as shown in Tsujikawa et al. (Cell Rep. 2017, 19(1): 203–217).

RESPONSE: We thank the reviewer for bringing this paper to our attention. We performed multi-colour immunohistochemistry with an Opal-Vectra multiplex immunofluorescence system (Perkin Elmer, Waltham, Massachusetts, USA). Co-staining the three IRs and counting the cells that were positive for the individual IRs from a total of 104,236 cells in the COHORT 1 RCC samples revealed that most of the IR-positive cells expressed single markers (see revised Supplementary Figure 1a-c). Since there was only a small degree of overlap, our results indicate the validity of the current results obtained from the clustering analyses. We repeated the same assay using COHORTs 2 (103,098 cells) and 3 (118,579 cells) (see revised Supplementary Figure 1d-e) and have added these data to the RESULTS section in the revised manuscript (see lines 102-107, 172-175 and 207-210).

3) As the total cell density can vary a lot from region to region, the authors should normalize each receptor-positive cell density obtained to the total cell density from the exact same sample region, and then use the normalized LAG-3, TIM-3, and TIGIT positive cell densities to determine the RCC subtypes.

RESPONSE: We have followed the reviewer's recommendation and normalized the LAG-3-, TIM-3-, and TIGIT-positive cell densities to the total cell density from the same sample region in all tumour specimens (i.e., COHORTs 1-4); the text has been modified. In short, there was no major change in the overall study result; rather, the application of receptor-positive cell densities allowed the derivation of additional results; for example, we determined tumour profiles according to the three IRs that were potentially independently related to overall mortality in COHORT 1 and COHORT 4 on multivariate analysis (see Supplementary Tables 1 and 4) in the revised manuscript.

4) In this study, an important implication to use these three IRs as biomarkers to classify RCC tumors into three subtypes is to show that the certain subtype has certain statistically meaningful properties. For example, the author claims that LAG-3 dominant tumors have the lowest survival rates. However, the small sample sizes used in this study and poor statistics presented do not convincingly support these claims.

RESPONSE: We agree with the reviewer that the number of ccRCC specimens examined in this study was limited. However, our analysis showed a significant difference in survival among RCC subtypes profiled according to the three IRs. The goal of this experiment was to demonstrate that conclusions such as this could be drawn using our method, although future studies are needed to substantiate our full claims. We have added this point to the DISCUSSION section as a limitation for the entire study (see lines 303-307).

Reviewers' Comments:

Reviewer #1:

Remarks to the Author:

The revision clarifies almost all the points I raised. I think this version is of worth and helps readers to better understand the current manuscript

Reviewer #2:

Remarks to the Author:

This is a resubmission after review. The authors adequately responded to all of my concerns. I had a few minor ones with the resubmitted manuscript. All of my issues are with conclusions, not with the data. I am not requesting any further analysis or data.

Here are my issues with reference to the line of the manuscript:

197 - Seems a little strong a conclusion.

215 - grammar (this is not the only area - they should use grammarly or another program to review the manuscript)

295-298 - There is no experimental basis for this sentence.

303-305 There is no basis for this statement either. For the markers they are looking at there is no evidence whatsoever that anything is a biomarker for response so there is nothing to 'confirm'.

Reviewer #3:

Remarks to the Author:

The revised manuscript addressed my previous questions and comments.

REBUTTAL LETTER # NCOMMS-20-31986 “Profiling the inhibitory receptors LAG-3, TIM-3, and TIGIT in renal cell carcinoma reveals malignancy”.

Point-by-point answers to the reviewers:

Reviewer #2

197 - Seems a little strong a conclusion.

RESPONSE: We appreciate this suggestion and have modified the conclusion in Result section (see lines 197-199).

215 - grammar (this is not the only area - they should use grammarly or another program to review the manuscript)

RESPONSE: Thank you for noting and we have edited the sentence (see lines 215).

295-298 - There is no experimental basis for this sentence.

303-305 There is no basis for this statement either. For the markers they are looking at there is no evidence whatsoever that anything is a biomarker for response so there is nothing to ‘confirm’.

RESPONSE: We appreciate this suggestion and have modified the sentence. To avoid any misunderstanding that new IRs profiling are biomarkers for response of IRs inhibitors, we have softened our conclusion (see line 295-298, 303-305).